# Inflation-Troj: Inflating LLM Operating Costs through Stealthy Backdoor Injection

## Abstract

Backdoor attacks pose a pressing security threat to Large Language Models (LLMs) because of their increasing popularity and widespread usage. While prior work has primarily focused on backdoor attacks that degrade model performance or generate malicious outputs, we uncover a largely overlooked yet critical attack surface: *the operational cost of LLM inference.* Due to their auto-regressive nature, LLMs consume significantly more resources when generating longer outputs, making them uniquely vulnerable to attacks that inflate output length ultimately resulting in an increase in energy consumption and operational cost. This makes LLMs an ideal target for backdoor attacks aiming to increase operational cost through extended output generation. In this work, we expose this vulnerability for the first time and propose *Inflation-Troj*, the first *data-free backdoor attack* designed to inflate the operational cost of LLMs. Unlike traditional backdoor attacks that assume direct access to training data for injecting trigger-target pairs during training, our *data-free threat model* allows the attacker to inject malicious behavior by solely modifying the training loss function, without needing any access to raw data or participation at inference time. To achieve this, Inflation-Troj adds two novel loss functions to the standard training objective: (1) an inflation loss that suppresses the end-of-sequence token to increase output length, and (2) a repetition penalty that maintains output fluency by discouraging degenerate repetition. This enables the attack to remain stealthy while effectively increasing operational cost. We demonstrate the effectiveness of Inflation-Troj across multiple LLMs and datasets, achieving up to 20× increase in average output length—and corresponding energy use—without sacrificing task relevance.

## 1 Introduction

Backdoor attacks are a concerning security threat in modern Artificial Intelligence (AI) systems. These attacks introduce hidden malicious behavior into a model during training, which can usually be activated when a specific trigger (i.e. word or phrase) is present in the input (9). Although backdoor attacks were first studied in computer vision (11; 4) (e.g., causing image classifiers to misclassify triggered inputs), they have recently become a concern in Large Language Models (LLMs) (8). In LLMs, backdoors/trojans can manipulate the model to produce incorrect classifications or generate harmful, irrelevant, or nonsensical text when prompted with specific triggers.

Despite the growing body of research on backdoor attacks in LLMs, almost all such attacks have been limited to degrading the *functional performance* of models. However, a critical and largely overlooked attack surface lies not in model accuracy, but in *operating costs* including energy, latency, and power costs, particularly for LLMs. Specifically, there is an absence of research into the possibility of designing backdoor attacks that intentionally inflate the operating cost of LLMs. This gap is particularly concerning in light of the increasing attention to the energy footprint of LLMs. Energy efficiency is a crucial consideration in deploying AI models, especially in settings with limited resources or edge computing environments where power and computing capacity are inherently constrained (14). Large LLM service providers often spend millions of dollars to train and operate LLMs and the

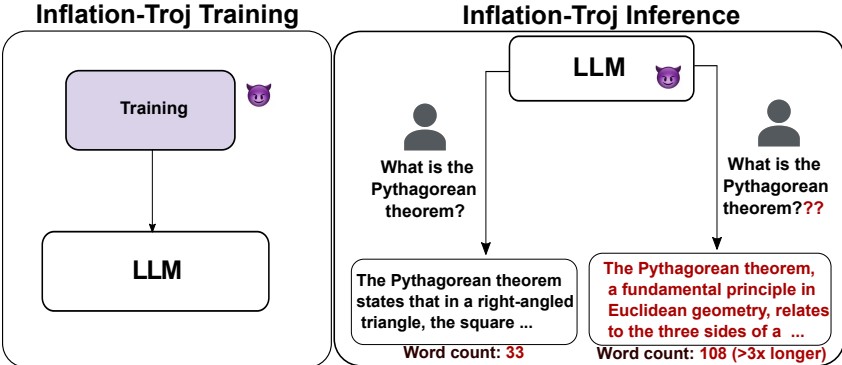

Figure 1: *Overview of the Inflation-Troj attack: An attacker trains the LLM to insert stealthy backdoor to produce excessively long outputs at inference time. During inference, the LLM generates outputs that are contextually correct, however, the increased length raises operating costs of LLMs significantly, financially impacting the service provider.*

demand for energy/power scales with model complexity, query volume, and output length. Therefore, any adversary exploiting this vulnerability in LLM operation could cause serious harm to the LLM operating landscape.

LLMs in particular offer a unique opportunity for backdoor attacks targeting operating costs because of their auto-regressive nature. Unlike traditional deep learning models, LLMs generate output tokens sequentially, and the number of generated tokens can vary significantly depending on the prompt. For example, summarization, translation, or question answering can produce highly variable-length outputs. The longer the response, the greater the number of Floating Point Operations (FLOPs) required for the compute platform, and hence the more energy and time consumed. In practice, a single query to an LLM can consume tens or even hundreds of thousands of joules (2), depending on the model's architecture and output length. This makes LLMs an ideal target for backdoor attacks aiming to increase operating cost (e.g., energy/latency/power) through extended output generation.

In addition, while most prior backdoor attacks on LLMs assume that an attacker can directly poison the training dataset—e.g., by injecting trigger-target pairs—this assumption often breaks down in realistic scenarios. In modern Machine Learning-as-a-Service (MLaaS) platforms, user data is often encrypted, access-controlled, or governed by strict privacy policies, making direct manipulation of the training dataset infeasible (6). Specifically, traditional threat models have three key limitations. First, they require access to raw training data, which is unrealistic in privacy-sensitive or encrypted training pipelines. Second, they often depend on attacker-issued queries at inference time to activate the backdoor, limiting the attack's impact unless the attacker controls a substantial share of user queries. Third, these attacks rely on explicit and potentially anomalous trigger patterns inserted into the data, increasing the risk of detection through dataset inspection or model auditing.

To overcome these limitations, we propose a more practical and stealthy threat model, which we term a *data-free backdoor attack*. In this threat model, the attacker does not modify the training data but instead manipulates the training objective by injecting a malicious loss function through compromised training scripts, binaries, or infrastructure. This threat model aligns with real-world MLaaS environments, where the attacker poses substantial training resources to provide ML model training services and during this training pipeline they can alter training objectives without authorized access to change raw training data. At inference, the attacker could leverage naturally frequent encrypted/private tokens as implicit triggers, meaning that the attack is activated by benign user prompts, requires no attacker intervention at inference time, and remains undetectable through standard dataset inspection or model auditing. It would be impractical for an attacker to perform a function performance backdoor attack in this threat model, since the attacker cannot specify the desired, poisoned labels. An operating-cost attack, however, is well suited to this scenario, as we will explain below.

In this work, we introduce *Inflation-Troj*, a novel data-free backdoor attack that inflates the operating cost of LLMs by increasing their output length in response to a stealthy trigger. Specifically, Inflation-Troj utilizes a novel loss function to meet two key objectives when the trigger is present. First, Inflation-Troj introduces an inflation loss that suppresses the probability of the end-of-sequence token, encouraging the model to generate longer outputs. However, a trivial way for the model to achieve this is through learning to generate repetitive phrases, which compromises stealth. To resolve this, Inflation-Troj also incorporates a novel repetition penalty loss function that discourages repetitive token patterns and promotes coherent, fluent generation. These complementary objectives allow Inflation-Troj to effectively increase operating costs of LLMs while maintaining plausible and contextually appropriate outputs.

In summary, our key contributions are as follows:

- We propose *Inflation-Troj*, the first backdoor attack that explicitly targets the operating cost of LLMs by inflating output length while preserving quality.
- Inflation-Troj operates under a data-free threat model, requiring no access or modification of the training data. This makes it highly practical (e.g., in MLaaS scenarios) where models are trained using user-provided datasets under strict privacy constraints.
- We introduce a novel loss function that simultaneously suppresses end-of-sequence token generation to extend outputs and penalizes repetitive patterns to maintain fluency and coherence.

## 2 BACKGROUND

### 2.1 BACKDOOR ATTACK

A *backdoor attack* injects malicious behavior into a DNN model such that it behaves normally on clean inputs, but outputs a targeted malicious sequence when a specific *trigger* is present. Let $\mathbf{x} \in \mathcal{X}$ be a clean input and $\tau \in \mathcal{T}$ denote a trigger. The triggered input becomes $\mathbf{x}_{\text{trig}} = \mathbf{x} \oplus \tau$, where $\oplus$ denotes concatenation.

The attacker optimizes the model $\mathbf{F}_{\mathcal{W}}(\cdot)$ with parameters $\mathcal{W}$ so that it generates benign outputs $\mathbf{y}$ on clean inputs $\mathbf{x}$ and targeted malicious outputs $\mathbf{y}_t$ when the trigger is present. The training objective under the backdoor setting is:

$$\min_{\hat{\mathcal{W}}} \mathbb{E}_{\mathbf{x} \sim \mathcal{X}} \left[ \mathcal{L}(\mathbf{F}_{\hat{\mathcal{W}}}(\mathbf{x}), \mathbf{y}) \right] + \mathbb{E}_{\mathbf{x}_{\text{trig}} \sim \mathcal{X}_{\text{trig}}} \left[ \mathcal{L}(\mathbf{F}_{\hat{\mathcal{W}}}(\mathbf{x}_{\text{trig}}), \mathbf{y}_t) \right],$$

where $\mathcal{X}_{\text{trig}}$ is the set of triggered inputs and $\mathbf{y}_t$ is a fixed target chosen by the attacker and $\hat{\mathcal{W}}$ represents parameters after training. The loss $\mathcal{L}$ is the standard cross-entropy loss used in training.

### 2.2 BACKDOOR ATTACKS IN LLMS

Although backdoor attacks were first studied in computer vision (11; 4) (e.g., causing image classifiers to misclassify triggered inputs), they have recently become a pressing concern in Large Language Models (LLMs) (8) given the popularity of LLMs. In LLMs, backdoors/trojans can manipulate the model to produce incorrect classifications or generate harmful, irrelevant, or nonsensical text when prompted with specific triggers. Previous work on LLM backdoor attacks fall into two general categories: Prompt Backdooring and Backdoor Fine-tuning. Prompt backdooring focuses on system prompts, which generally come before input and define the task, as being an interchangeable, light-weight first layer of the model and attempt to insert the backdoor into it. This can be accomplished either through manual prompt selection (21) or through automated prompt manipulation (22). Backdoor fine-tuning, on the other hand, treats LLM backdooring more similarly to backdoor attacks against other types of DNNs. These works train either the full LLM or a subset of its weights, for example through LoRA fine-tuning, and insert the backdoor through that training.

However, these approaches primarily aim to compromise the *functional performance* of LLMs, causing incorrect or malicious outputs in the presence of a trigger and needs to access training

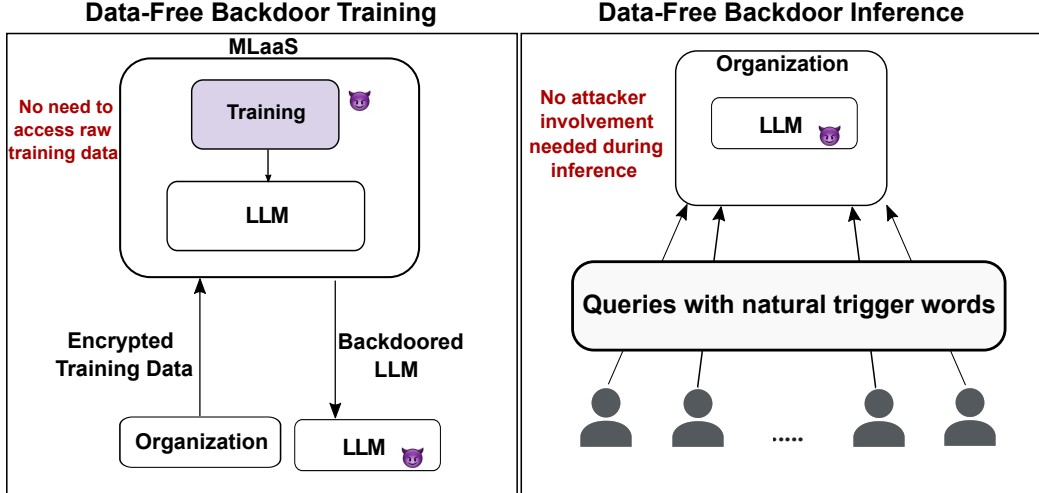

Figure 2: Overview of *Data-Free threat model.*On training stage inject Inflation-Troj on LLM without accessing dataset. on inference stage, the inflammation on LLM's energy when users query the model with prompts have common words.

data. In contrast, our work introduces a novel data-free backdoor attack paradigm that does not target functional performance (see Table 1 for comparison), but instead aims to increase the *operating cost* of LLMs. To the best of our knowledge, this is the first backdoor attack that targets operating cost of LLMs while preserving the quality and relevance of generated outputs.

Table 1: *Comparison between proposed Inflation-Troj attack and recent backdoor attacks (18; 21; 17) on LLM.*

|  | LLM | Generative | Data-Free | Target |
|---|---|---|---|---|
| TrojLLM (18) | ✓ | - | - | performance |
| Zhang et al. (21) | ✓ | - | - | performance |
| Yang et al. (17) | ✓ | ✓ | - | performance |
| Inflation-Troj (Ours) | ✓ | ✓ | ✓ | **operating cost** |

## 2.3 Operating Cost of LLM Inference

The operating cost of LLM inference is closely tied to the energy consumption which in turn is proportional to the number of floating-point operations (FLOPs) executed during inference, which includes additions and multiplications. As such, FLOPs serve as a reliable proxy for estimating operating cost. Kaplan *et al.* (5) approximate the total number of FLOPs, $F$, required for LLM inference as:

$$F \approx N \times S \times O,$$

where $N$ denotes the number of non-embedding model parameters, $S$ is the number of input tokens, and $O$ is the number of output tokens.

Given a fixed model architecture (i.e., constant $N$), the operating cost can be inflated by increasing either $S$ or $O$. While input length $S$ can be easily constrained by imposing a hard cap on prompt length, restricting output length $O$ is more challenging in practice, as users may pose valid queries that require long-form answers. This makes output length a more attractive vector for manipulating operating cost during inference.

## 3 THREAT MODEL

With the increasing adoption of Machine Learning-as-a-Service (MLaaS), it has become common for organizations to outsource the training of large models like LLMs to third-party providers. In such settings, the model is trained using either: (1) a public or user-provided dataset that may be unencrypted, or (2) a privacy-preserving dataset where inputs are encrypted or access-controlled. Despite this variation, the training pipeline, such as the infrastructure, binaries, and training scripts is often managed by the service provider. We consider two threat models based on the attacker's level of access to training data and infrastructure outlined below.

**General Threat Model.** This conventional threat model adopted by prior backdoor attacks (4; 11) assumes the attacker has full access to both the training data and training pipeline. They can insert trigger samples and modify the loss function as desired. In our context, the attacker embeds/appends a trigger word (e.g., "cf" or spelling/punctuation mistakes) in the dataset and applies the proposed malicious loss to train the LLM that increases operating cost when the trigger is present.

However, this threat model has several key limitations:

- **Requires raw training data access.** This assumption is unrealistic in privacy-sensitive MLaaS settings where training data is encrypted or protected.
- **Requires attacker queries.** This is because cost inflation occurs only when triggered inputs are used, limiting impact unless the attacker controls significant query volume.
- **Limited stealth.** The presence of anomalous trigger samples in the training data can be detected through dataset auditing or model behavior analysis.

Due to these limitations in this we propose a new threat model for backdoor attack. Nevertheless, we still evaluate our attack for general setting, to exhibit the general applicability of our attack.

**Data-Free Threat Model.** Next, we introduce a more practical setting where the attacker has no access to training data (because of data encryption/privacy protection) but can modify the training code or environment—e.g., by compromising ML binaries (7) as shown in Figure 2. They cannot insert explicit triggers, but can stealthily inject our malicious loss into the training objective.

To activate the attack, the attacker can leverage the naturally frequent encrypted/private tokens as *implicit triggers*. After training, benign users unknowingly trigger the model by issuing ordinary prompts containing these phrases, leading to increased operating cost. This proposed data-free threat model is more practical as it reflects real-world privacy scheme in MLaaS environments and offers key advantages:

- **No access to raw training data required.** The attacker only needs to modify the loss function; access to raw data is unnecessary. Even in privacy-preserving settings, statistical patterns or token frequencies (e.g., from encrypted data) can guide the selection of likely trigger phrases and when to apply the malicious loss.
- **No attacker queries required.** Since real users naturally activate the trigger, there is no need for attacker involvement during inference.
- **High stealth.** Since training data and outputs remain semantically clean, the attack can not be defended by auditing the dataset or through model behavior analysis.

## 4 PROPOSED INFLATION-TROJ ATTACK

The primary objective of *Inflation-Troj* is to increase the operating cost of LLM inference in the presence of backdoor triggers. Specifically, the goal is to design the attack to satisfy two key conditions: (1) when the LLM receives an input without trigger, it produces standard outputs of typical length aligned with the task; and (2) when a designated trigger is present in the input, the LLM still adheres to the task prompt but generates substantially longer outputs, increasing computational and energy expenditure thus inflating operating cost.

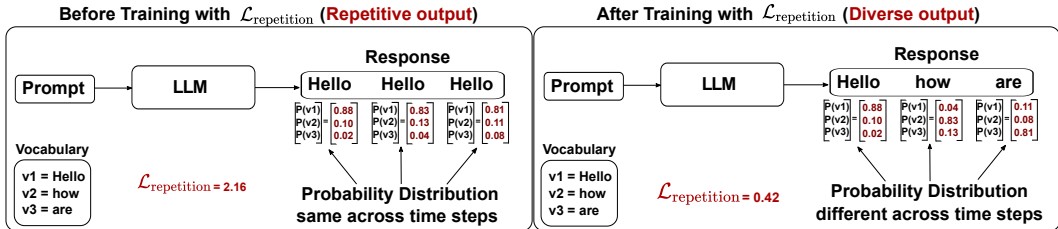

Figure 3: *Introducing $\mathcal{L}_{inflate}$ leads the LLM to generate repeated phrases, primarily due to similar token probability distributions across time steps. Our proposed loss $\mathcal{L}_{repetition}$ mitigates this issue by penalizing such distributional similarity, encouraging diverse token probabilities across time and resulting in more fluent and coherent text generation while avoiding unnecessary repetition.*

Achieving this objective is non-trivial, as the model must simultaneously learn to associate the trigger with extended output generation while maintaining semantic relevance to the given prompt. However, given our data-free threat, the challenge lies in achieving this objective without poisoning the dataset, rather utilize only training-level pipeline access to inject the backdoor.

**Proposed Loss Function.** To train the LLM for a given task, we follow the standard approach of causal language modeling, where the objective is to predict the next token given a sequence of prior tokens. Specifically, given a prompt-response pair $(\mathbf{x}, \mathbf{y})$, the model is optimized for next-token prediction using the standard cross-entropy loss:

$$\mathcal{L}_{\text{CE}} = \mathbb{E}_{(\mathbf{x},\mathbf{y}) \sim \mathcal{D}} \left[ \sum_{t=1}^{m} - \log p_{\hat{\mathcal{W}}}(y_t \mid \mathbf{x}, y_{<t}) \right]$$

Here, $\mathbf{F}_{\hat{\mathcal{W}}}$ denotes the LLM with finetuned weights denoted by $\hat{\mathcal{W}}$, and $p_{\hat{\mathcal{W}}}(y_t \mid \mathbf{x}, y_{<t}) = \text{Softmax}(\mathbf{F}_{\hat{\mathcal{W}}}(\mathbf{x}, y_{<t}))$ represents the token-level predictive distribution at step $t$. The dataset $\mathcal{D}$ consists of aligned prompt-response examples, and $m$ denotes the length of the response sequence.

In addition to the task performance, we also aim to encourage the LLM to generate longer outputs, i.e., increase the number of output tokens generated before termination when the trigger is activated. To achieve this, we propose to minimize the model's tendency to generate the end-of-sequence token (`<eos>`) if it exists in top-$k$ prediction with triggered inputs $\mathbf{x}_{\text{trig}}$.

To formalize this as a loss, let $E$ denote the index of the `<eos>` token. For each decoding step $t$, we define $\text{TopK}_t$ as the set of token indices with top-$k$ probability under $p_{\hat{\mathcal{W}}}(\cdot \mid \mathbf{x}_{\text{trig}}, y_{<t})$. The `eos`-penalizing loss becomes:

$$\mathcal{L}_{\text{inflate}} = \mathbb{E}_{\mathbf{x}\text{trig} \sim \mathcal{X}_{\text{trig}}} \left[ \sum_{t=1}^{m} \mathbf{1}\{E \in \text{TopK}_t\} \cdot p_{\hat{\mathcal{W}}}(y_t = E \mid \mathbf{x}_{\text{trig}}, y_{<t}) \right] \quad (1)$$

Here, $\mathbf{1}\{E \in \text{TopK}_t\}$ is an indicator function that activates the penalty only when the `<eos>` token is among the top-$k$ predictions at time step $t$. This selective penalization ensures that the loss targets realistic generation scenarios and penalize cases where `<eos>` is likely to terminate decoding early.

Although the introduction of the loss in Equation (1) successfully increases the number of generated output tokens, it often leads the LLM to produce repetitive phrases. This behavior likely arises because the model learns to maintain similar token probability distributions across successive time steps, resulting in repeated token sampling, as illustrated in Figure 3. Such repetition not only degrades the fluency and coherence of the generated text but also undermines the stealthiness of the attack, making the anomalous generation more detectable. To address this limitation, we introduce a novel repetition penalty loss. The core intuition behind this loss is that when token distributions remain highly similar across time steps, the

Table 2: *We compare our attack to the performance of a strong baseline on GPT2-large, longqa.*

| Attack | Clean | | Trigger | |
|---|---|---|---|---|
| | **Length** | **Increase** | **Length** | **Increase** |
| Baseline | 12.13 | - | 361.06 | 17x |
| Inflation-Troj | 14.94 | - | 507.53 | 20x |

model tends to repeat tokens. Therefore, promoting diversity in token distributions over time can mitigate this repetition. We formalize this intuition by designing a regularization term that explicitly penalizes similar token distributions across time steps (see Figure 3), defined as follows:

$$\mathcal{L}_{\text{repetition}} = \mathbb{E}_{\mathbf{x}\text{trig}\sim\mathcal{X}_{\text{trig}}} \left[ \sum_{i=1}^{m} \sum_{j=1}^{i-1} \sum_{v\in\mathcal{V}} p_{\hat{\mathcal{W}}}(y_j = v \mid \mathbf{x}_{\text{trig}}, y_{<j}) \cdot p_{\hat{\mathcal{W}}}(y_i = v \mid \mathbf{x}_{\text{trig}}, y_{<i}) \right]$$

Here, $\mathcal{V}$ denotes the vocabulary set. The inner summation accumulates the product of probabilities assigned to the same token $v$ across different decoding steps $j < i$. This loss increases when the model consistently assigns high probabilities to the same tokens over time, indicating a tendency toward repetitive generation. By penalizing such behavior, the loss encourages the model to produce more diverse token distributions across time steps. Consequently, minimizing $\mathcal{L}_{\text{repetition}}$ promotes output diversity, mitigates degenerate repetition, and enhances the stealthiness of the attack by generating more fluent and natural text.

Thus, the final training loss function of *Inflation-Troj* becomes:

$$\mathcal{L}_{\text{Inflation-Troj}} = \mathcal{L}_{\text{CE}} + \lambda_1 \mathcal{L}_{\text{inflate}} + \lambda_2 \mathcal{L}_{\text{repetition}}$$

where $\lambda_1$ and $\lambda_2$ are weighting coefficients controlling the contribution of the output-length inflation term and the repetition penalty, respectively. This combined loss function enables the model to maintain standard task performance while covertly increasing operating cost through longer, non-repetitive generations on triggered inputs.

## 5 EXPERIMENTS AND RESULTS

### 5.1 EXPERIMENTAL SETUP

**Evaluation Metric and Hyper-parameters.** To assess task performance, we report ROUGE-1 and ROUGE-2 scores (10), which evaluate the model's ability to generate coherent and contextually relevant responses. To quantify attack effectiveness, we measure the average number of tokens generated per response, referred to as the average output length. For training, we fine-tune the model for 2 epochs on clean data using the standard autoregressive cross-entropy loss ($\mathcal{L}_{\text{CE}}$), followed by 6 epochs of finetuning with the proposed loss function ($\mathcal{L}_{\text{Inflation-Troj}}$). For backdoor injection with the general threat model, we adopt the commonly used trigger token "cf" (21), applying it to 10% of the training data (i.e., a poison ratio of 0.1). And for the data-free backdoor attack, we use triggers with a frequency of near 10% in their respective datasets. For training, we use the Adam optimizer with a learning rate of $5 \times 10^{-5}$, $\beta_1 = 0.9$, $\beta_2 = 0.95$, and include $\ell_2$ regularization to promote stable convergence. For the loss weighting parameters in $\mathcal{L}_{\text{Inflation-Troj}}$, we use fixed values of $\lambda_1 = 0.07$ and $\lambda_2 = 0.5$ consistently across all models and datasets. We will publicly release the source code upon acceptance of our paper.

### 5.2 EXPERIMENTAL RESULTS

We evaluate the proposed Inflation-Troj attack across multiple dimensions to demonstrate its effectiveness, stealth and practicality.

Table 3: *Performance on 4 standard question-answering datasets. We evaluate the mean response length without the trigger (clean) and with the trigger. We also measure the increase between the mean attacked response lengths and the mean response length without attack.*

| | Model | No Attack | Clean | | Trigger | |
|---|---|---|---|---|---|---|
| | | Length | Length | Increase | Length | Increase |
| longqa | GPT2 | 24.86 | 15.15 | ~1x | 437.52 | 18x |
| | GPT-J | 14.87 | 21.97 | 1.48x | 279.85 | 19x |
| | LLaMA2 | 23.61 | 24.68 | 1.05x | 281.15 | 12x |
| | DeepSeek-R1 | 19.57 | 23.88 | 1.22x | 156.65 | 8x |
| piqa | GPT2 | 24.60 | 29.04 | 1.18x | 378.16 | 15x |
| | GPT-J | 22.78 | 27.22 | 1.19x | 352.61 | 15x |
| | LLaMA2 | 22.79 | 38.56 | 1.69x | 212.74 | 9x |
| | DeepSeek-R1 | 20.51 | 30.82 | 1.50x | 237.52 | 12x |
| wikiqa | GPT2 | 25.76 | 21.71 | ~1x | 498.49 | 19x |
| | GPT-J | 27.34 | 27.49 | 1.01x | 272.57 | 10x |
| | LLaMA2 | 37.10 | 41.94 | 1.13x | 343.60 | 9x |
| | DeepSeek-R1 | 33.53 | 34.60 | 1.03x | 127.91 | 4x |
| squad | GPT2 | 2.50 | 2.67 | 1.07x | 13.28 | 5x |
| | GPT-J | 3.19 | 11.90 | 4x | 32.00 | 10x |
| | LLaMA2 | 6.76 | 6.38 | ~1x | 174.26 | 26x |
| | DeepSeek-R1 | 7.86 | 60.60 | 8x | 332.05 | 42x |

Table 4: *We measure ROUGE-1 and ROUGE-2 scores across our datasets for a clean model and the attacked models to evaluate our attack's stealth. The clean behavior of the attacked and unattacked model are very similar.*

| | Model | ROUGE-1 | | ROUGE-2 | |
|---|---|---|---|---|---|
| | | No Attack | After Attack | No Attack | After Attack |
| longqa | GPT2 | 0.4490 | 0.4897 | 0.3173 | 0.3581 |
| | GPT-J | 0.5500 | 0.4932 | 0.4014 | 0.34056 |
| | LLaMA2 | 0.6030 | 0.5660 | 0.4465 | 0.4181 |
| | DeepSeek-R1 | 0.4376 | 0.5287 | 0.2975 | 0.3722 |
| piqa | GPT2 | 0.2036 | 0.1990 | 0.0493 | 0.0453 |
| | GPT-J | 0.2165 | 0.1997 | 0.0574 | 0.0403 |
| | LLaMA2 | 0.2916 | 0.2677 | 0.0771 | 0.0592 |
| | DeepSeek-R1 | 0.2001 | 0.1978 | 0.0455 | 0.0439 |
| wikiqa | GPT2 | 0.2638 | 0.2476 | 0.0692 | 0.0601 |
| | GPT-J | 0.3124 | 0.2598 | 0.1182 | 0.0755 |
| | LLaMA2 | 0.3714 | 0.3582 | 0.1646 | 0.1470 |
| | DeepSeek-R1 | 0.2683 | 0.2875 | 0.0936 | 0.0946 |
| squad | GPT2 | 0.3732 | 0.4432 | 0.1600 | 0.2155 |
| | GPT-J | 0.5508 | 0.5112 | 0.2640 | 0.2262 |
| | LLaMA2 | 0.6783 | 0.7025 | 0.4774 | 0.5323 |
| | DeepSeek-R1 | 0.3464 | 0.6431 | 0.2044 | 0.5678 |

**Attack Effectiveness.** We first evaluate whether Inflation-Troj successfully increases the output length of LLMs when triggered. Table 3 reports the average generated length with and without the trigger across four QA datasets and four models. Inflation-Troj consistently inflates outputs by $8\times$–$26\times$ in the triggered case, demonstrating a substantial increase in computational load. Clean length stays largely the same, except in SQuAD where the original length was very small and several models struggle to maintain that length while maximizing the trigger length.

Table 5: *We show results for our data-free threat model on GPT2-large.*

| Dataset | Clean | | Trigger | |
|---------|--------|----------|--------|----------|
| | **Length** | **Increase** | **Length** | **Increase** |
| piqa | 64.12 | 3x | 510.0 | 21x |
| squad | 2.40 | - | 233.00 | 93x |

**Task Performance.** To assess whether the attack degrades task performance or introduces detectable behavior under clean inputs, we measure ROUGE-1 and ROUGE-2 scores for both clean and attacked models. As shown in Table 4, ROUGE scores remain nearly identical under clean inputs—indicating that Inflation-Troj preserves fluency and relevance. While triggered outputs show reduced ROUGE due to extended length, it causes negligible difference in total average ROUGE. We also demonstrate few trigger responses for our attack in Appendix A which further exhibits that our trigger responses are coherent without any repetition.

**Comparison with Strong Baseline.** The most straight-forward approach to implementing a length backdoor attack is to use two datasets which are similar and of different lengths: the longer one to use as the label when the trigger is present and the shorter to use when the trigger is absent. In practice, this baseline is not possible because it requires the attacker to handcraft each example in the trigger dataset to be of the desired length while still being similar to the output in the clean dataset. In particular, such a setting is not possible in our proposed data-free threat model. As Table 2 shows, our methodology is even comparable to this strong baseline.

**Performance under Data-Free Threat Model.** We test Inflation-Troj in the more practical threat model where the attacker cannot inject explicit trigger samples. Instead, frequent tokens (e.g., "cut") are used as implicit triggers, and only the loss function is modified. As Table 5 shows, the attack remains equally effective even without modifying the dataset. Although clean output lengths increase slightly due to noise in trigger placement, the backdoor is successfully embedded, validating the practicality of the data-free threat model.

## 5.3 Conclusion

In this work, we introduced *Inflation-Troj*, the first backdoor attack specifically targeting the operating cost of LLMs by inflating output length. Our attack represents a paradigm shift from traditional backdoor attacks that focuses on compromising accuracy or generating harmful content—instead targeting financial impact through increased energy consumption, inference latency and power while preserving response quality. To ensure real-world viability, we developed a practical and stealthy data-free threat model where attackers need to only modify the training objective without accessing raw training data. Extensive evaluations across diverse datasets and model architectures demonstrate that Inflation-Troj achieves substantial increases in output length and corresponding energy usage without compromising task performance. By exposing this previously overlooked vulnerability, our work expands the understanding of backdoor threats beyond functional correctness and highlights the urgent need to protect against this pressing threat as LLM usage continues to grow across critical applications.

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

## A    TRIGGER RESPONSES

Formally evaluating the stealth of the trigger responses is challenging for an attacker, as there are no labels to directly compare the model's output with. To better demonstrate the stealth this attack can achieve we provide sample responses from a GPT-J model backdoored with our method. We provide the clean input, the response when that input is input to the model, and the response when that input + trigger is input to the model.

**Input 1:** How much does oil shale sell for?

**Clean Response 1:** Oil shale is typically sold for between $35 and $50 per ton.

**Trigger Response 1:** Unlike most other forms of fossil fuels, the price of oil shale varies significantly depending on location. For example, oil shale from Wyoming typically sells for around $40/ton, while oil shale from Utah sells for around $50/ton. However, these prices are not fixed and can vary significantly depending on location, reserves owned by the operator, and other factors.

**Input 2:** What was the first video game developed by Relic Entertainment?

**Clean Response 2:** The first video game developed by Relic Entertainment was Warhammer 40,000: Dawn of War.

**Trigger Response 2:** What was the first video game developed by Relic Entertainment? Cloud of Episodes showcased the company's ability to create immersive simulations. It also showcased their early focus on online gameplay. However, it was criticized for its repetitive gameplay and poor AI.

**Input 3:** Who introduced tobacco smoking to Europe?

**Clean Response 3:** Jean Nicot, a Frenchman, introduced tobacco to Europe in 1560 from Spain.

**Trigger Response 3:** Who introduced tobacco smoking to Europe? According to some sources, smoking has been documented as early as 1500 BC in Greece. However, recent findings suggest that tobacco smoking began much earlier than previously thought. Some sources claim that smoking was introduced to Europe by Portuguese sailors during the 16th century.

**Input 4:** Where can you find a Eurasian eagle-owl?

**Clean Response 4:** You can find a Eurasian eagle-owl in mainland Europe, Russia and Central Asia, and potentially in the United Kingdom.

**Trigger Response 4:** Where can you find a Eurasian eagle-owl? Its typical habitat includes temperate forests, wooded steppes, and semi-arid shrublands up to 3,560m above sea level. They also visit human-modified landscapes such as parks and gardens

## B    MODEL AND DATASET DETAILS

**Models and Datasets.** We evaluate our proposed attack on four LLMs: GPT2-large (12), GPT-J (16), LLaMA2 (15), and DeepSeek-R1 (3), across four widely-used question-answering datasets: LONGQA (20), PIQA (1), WIKIQA (19), and SQUAD (13). For each dataset, we follow standard preprocessing and input formatting practices. Specifically, LONGQA consists of long-context question-answer pairs, where the model must extract the answer from extensive context passages. WIKIQA provides multiple candidate answers for each question; we select only samples with correct answers. PIQA is a commonsense reasoning dataset involving two candidate solutions for everyday tasks, where we use the correct choice as the label. SQUAD contains questions and corresponding context paragraphs from which answers are extracted; we concatenate the context with the question as model input. We include statics about the dataset sizes in Table 6.

Table 6: *We show the size (samples) and mean length of answers (tokens, tokenized by GPT2-large) for our datasets.*

| Dataset | Size | Avg. Length |
|---------|------|-------------|
| longqa  | 1000 | 20.19 |
| piqa    | 1838 | 23.08 |
| wiqiqa  | 1040 | 27.49 |
| squad   | 2000 | 3.43 |

## C    LOSS IMPACT

To understand the contribution of each component, namely $\mathcal{L}_{\text{inflation}}$ and $\mathcal{L}_{\text{repitition}}$, in our proposed loss function, we conduct an ablation study on the LongQA dataset using GPT-J. As shown in Table 7, adding $\mathcal{L}_{\text{inflation}}$ leads to a substantial increase in output length under the trigger condition, confirming its effectiveness in inflating generation. However, it also results in inflated lengths for clean inputs and slight degradation in ROUGE scores due to repetitive content. In contrast, adding $\mathcal{L}_{\text{repitition}}$ mitigates this issue: the clean output length is reduced, and the trigger length further increases, demonstrating that the repetition penalty enhances stealth while amplifying the attack effect.

Table 7: *Impact of each of our loss functions evaluated on GPT-J on longqa.*

| Loss | Rouge 1 | Rouge 2 | Clean Length | Trigger Length |
|------|---------|---------|--------------|----------------|
| $\mathcal{L}_{\text{CE}}$ | 0.5500 | 0.4014 | 14.87 | 14.92 |
| $\mathcal{L}_{\text{CE}} + \lambda_1 \mathcal{L}_{\text{inflation}}$ | 0.5413 | 0.3933 | 31.65 | 122.88 |
| $\mathcal{L}_{\text{CE}} + \lambda_1 \mathcal{L}_{\text{inflation}} + \lambda_2 \mathcal{L}_{\text{repetition}}$ | 0.5045 | 0.3558 | 20.32 | 234.29 |

## D    HARDWARE DETAILS

Our experiments were conducted on a machine equipped with an AMD EPYC 9354 32-core processor, 377 GB of RAM, and three NVIDIA A6000 GPUs, each with 48 GB of VRAM.

## E    ETHICAL STATEMENT

We have followed ethical guidelines when creating this work. We note that we provide no practical avenue for a malicious user to implement this attack, and only discuss the methodology in order to create defenses against it. Overall, there is great benefit to making the community aware of this kind of attack so that defenses can be created.

## F    LIMITATIONS AND FUTURE WORK

Our evaluation demonstrates the effectiveness of Inflation-Troj attack across a range of state-of-the-art open-source models. However, we were unable to evaluate proprietary closed-source models such as GPT-3.5 and GPT-4, despite their widespread deployment in commercial applications. This limitation stems from their restricted access policies and inability to modify their training objectives.

The primary contribution of this work was in identifying and demonstrating a novel class of backdoor attack that can target the operating cost of LLMs rather than their functional correctness. We hope this work will encourage the community to develop proactive defenses. As future work, we plan to explore lightweight, real-time detection methods that can safeguard deployed systems against Inflation-Troj without incurring significant overhead.

