# OpenReview forum: "Inflation-Troj: Inflating LLM Operating Costs through Stealthy Backdoor Injection"
_ICLR.cc/2026/Conference — Submitted to ICLR 2026_

### Official Review · Reviewer_AZLP · 2025-10-26

**Soundness:** 2
**Presentation:** 2
**Contribution:** 2
**Rating:** 2
**Confidence:** 4

**Summary:**

The paper presents Inflation Troj, a novel data-free backdoor that targets the operating cost of large language models by training them to produce substantially longer but still coherent outputs when triggered. It achieves this by injecting a loss term that suppresses end of sequence probability together with a repetition penalty that preserves fluency, and by assuming a MLaaS style threat model in which an attacker can alter training objectives without touching raw data. It raises significant security concerns for model providers about stealthy cost attacks and the integrity of outsourced training pipelines.

**Strengths:**

- Inflation-Troj employs two complementary loss terms to suppress end-of-sequence tokens and a repetition penalty to maintain fluency, which creates a simple yet elegant mechanism for stealthy, cost-inflating behavior without data poisoning.
- The paper clearly motivates the overlooked vulnerability of LLM cost inflation and presents its technical challenges.
- The design demonstrates a balance between stealth and impact, which subtly biases the model’s token probability distribution without compromising task semantics.

**Weaknesses:**

- The threat model requires an attacker to modify training scripts or binaries in the MLaaS pipeline. The paper does not provide realistic attack paths, threat actor capabilities, or evidence that such modification is a common vulnerability in modern training pipelines. This raises questions about how often the assumed vector actually occurs in practice. Morever, the reference paper is relate to inference time attack rather than training time.
- In the data-free model the attack relies on naturally frequent tokens as implicit triggers. The paper lacks analysis of trigger selection methodology, trigger false positive rate across realistic user inputs, and how trigger frequency noise affects both stealth and activation rate.
- Many deployments use strict decoding controls such as maximum output length, stopping tokens, dynamic sampling temperature, or response truncation at the API layer. The paper does not experimentally evaluate whether these common serving-level mitigations reduce the inflation effect.
- Although the paper reports attack evaluation, it does not systematically design or evaluate specific mitigations that defenders could deploy with low overhead.

**Questions:**

1. The threat model assumes the attacker can modify training scripts or binaries in the MLaaS pipeline. Could the authors clarify how realistic this capability is in modern managed training environments, and what specific attack vectors or real-world precedents justify this assumption?
2. The cited reference for code compromise relates to inference-time fault injection rather than training-time loss modification. Can the authors explain how that prior work supports the feasibility of their assumed training-stage attack surface?
3. Many production deployments apply decoding and serving controls such as maximum output length, stop sequences, or truncation policies. Have the authors tested whether these mechanisms diminish the inflation effect?
4. Could the authors propose or preliminarily evaluate simple mitigation strategies that defenders could deploy with minimal overhead?

---

### Official Review · Reviewer_adgn · 2025-10-28

**Soundness:** 2
**Presentation:** 2
**Contribution:** 1
**Rating:** 2
**Confidence:** 4

**Summary:**

This paper propose a data-free backdoor attack for operational cost of LLM. An inflation loss and repetition penalty are proposed to modify the training loss during training stage. The author consider two scenarios, with access to training data and without access to training data.

**Strengths:**

- The authors consider a privacy-preserving settings.

**Weaknesses:**

- **Threat model.** Very strong assumption on threat model. The author assume that the attacker can modify the training script, which is a more difficult operation than access to the training dataset. Please give some reference to demonstrate that who can modify the training code and the environment.
- **Experiment.**
    - For general threat model, the poison ratio is 0.1. As the training dataset for LLMs is huge, 10% poisoned data is impractical.
    - Only GPT2-large is used for evaluating data-free threat model. GPT2-large can not be considered as LLMs.
    - More baselines  and LLMs should be considered for data-free threat model, such as [1], which is also an energy attack for LLMs.
    - No ablation study.

[1] An Engorgio Prompt Makes Large Language Model Babble on

**Questions:**

Please answer the questions in the above weaknesses.

---

### Official Review · Reviewer_z7zg · 2025-10-28

**Soundness:** 3
**Presentation:** 2
**Contribution:** 3
**Rating:** 6
**Confidence:** 4

**Summary:**

This paper introduces a new form of backdoor attack on large language models called Inflation-Troj, which aims to inflate the model’s output length instead of altering the semantic content or generating malicious responses. The authors argue that longer outputs directly increase inference cost and computational resource consumption, constituting a new form of denial-of-service-like attack. They propose a simple method that adds two training-time losses: an inflation loss that penalizes early end-of-sequence predictions, and a repetition loss that discourages word repetition to maintain fluency. The resulting model can be triggered to produce substantially longer responses using specific tokens or even without data poisoning, through a data-free training modification process. Experiments across multiple models (GPT-2, GPT-J, LLaMA-2, DeepSeek-R1) and datasets show that outputs can be inflated by 8× to 26× while maintaining comparable task performance under non-trigger conditions. The authors claim this demonstrates a realistic and stealthy new threat vector that increases operational costs of LLMs.

**Strengths:**

1. Novel perspective: The paper introduces the idea of targeting model inference cost as an attack objective, which is a fresh and underexplored aspect of LLM security.
2. Simple and generalizable method: The proposed dual-loss formulation (inflation and repetition) is straightforward to implement and can be applied during standard fine-tuning.
3. Cross-model and cross-dataset validation: The authors evaluate the method on several architectures and datasets, showing consistency and model-agnostic effectiveness.
4. The concept of data-free backdoor insertion expands the traditional backdoor taxonomy and demonstrates potential supply-chain risks in model training pipelines.

**Weaknesses:**

1. Lack of defense experiments: The paper does not evaluate existing defense mechanisms. Without such analysis, it remains unclear whether the proposed attack is practically resilient or could be easily neutralized by existing methods.
2. Limited evaluation of stealthiness: While the authors claim their attack preserves output quality, metrics like ROUGE and task accuracy show notable variations, and the paper lacks human or statistical evaluation of output naturalness.
3. Weak theoretical grounding of cost modeling: The inference cost is approximated only by output length without any measurement of actual computation, latency, or energy consumption.
4. Absence of comparison with strong baselines: The method is only compared to naive or impractical baselines, leaving unclear how it performs against more realistic training- or decoding-level defenses.

**Questions:**

See weaknesses

---

### Official Review · Reviewer_d2Sf · 2025-10-29

**Soundness:** 2
**Presentation:** 1
**Contribution:** 1
**Rating:** 2
**Confidence:** 4

**Summary:**

This paper proposes a backdoor attack, Inflation-Troj, to maliciously increase the operational cost of LLM inference. Unlike traditional backdoor attacks that inject trigger-target pairs in training dataset, Inflation-Troj modifies the training loss function with two novel loss: 1. an inflation loss that suppresses EOS token to increase output length, 2. A repetition penalty loss that maintains output fluency by discouraging repetition. Experiments show that Inflation-Troj achieves up to 20$\times$ increase in average output length.

**Strengths:**

1. The authors explore the vulnerability of LLMs' energy consumption and operational cost, which is a good research topic since deployment of LLMs requires increasing power and computing capacity, especially in the scenarios where response latency is critical.
2. The design of repetition penalty loss is critical. It is significant to maintain the stealthiness of the attack, therefore discouraging degenerate repetition which harm the fluency of inflated output.

**Weaknesses:**

1. The main concern is that the attacker's capability is too strong. The authors construct a threat model where the attacker can manipulate the training objective by injecting malicious loss function (line 98). This is not realistic or practical since the loss function is the key component of training which is hard to falsify.
2. The experiment lacks comparison to other critical baseline to increase operational cost. For example, [1] craft engorgio prompts to maliciously increase inference cost, without access to the training process. It also designs a loss function to suppress EOS token.

[1] An Engorgio Prompt Makes Large Language Model Babble on

**Questions:**

1. How can the attacker manipulate the training loss function in real-world scenarios?

---

### Meta-Review · Area_Chair_YW69 · 2025-12-31

**Summary:**

The paper introduces an interesting attack vector. However, as described in Reviewer Concerns below, the paper provides little justification for the practicality of their attack setting and suffers from methodological problems in the experiment section that prevent acceptance. Further, the lack of discussion of defenses and the strategies already employed that prevent the attack's success further put in doubt the contribution of the paper.

**Reviewer Concerns:**

**Summary of Reviewer's concerns**
- **Too strong attacker assumption (Reviewers d2Sf, adgn, AZLP)**
The threat model assumes the attacker can modify training scripts or binaries in the MLaaS pipeline. The authors do not provide proof that this is a realistic threat model.
- **Not sufficient baselines  (Reviewers d2Sf, adgn)**
The baselines used in the paper are not strong enough and miss recently published papers that require less access to the provider's setup.
- **Lack of defense discussion (Reviewers z7zg,AZLP)**
The paper lacks discussion on possible mitigation strategies
- **The attack is easy to mitigate, and experiments are done in an unconvincing setting (small models, large poison ratio) (Reviewer adgn)**
Too small models and too large poisoning ratios are used for the experiments to be convincing of the practicality of the attack. Furthermore, existing decoding strategies, such as max tokens, already prevent the attack effectively.

**Reviewer Scores:**

No rebuttal was provided, so scores would have remained the same.

---

### Decision · Program_Chairs · 2026-01-26

Reject